# A Supervised Learning Regression Method for the Analysis of the Taste Functions of Healthy Controls and Patients with Chemosensory Loss

**DOI:** 10.3390/biomedicines11082133

**Published:** 2023-07-28

**Authors:** Lala Chaimae Naciri, Mariano Mastinu, Melania Melis, Tomer Green, Anne Wolf, Thomas Hummel, Iole Tomassini Barbarossa

**Affiliations:** 1Department of Biomedical Sciences, University of Cagliari, 09042 Cagliari, Italy; l.naciri@studenti.unica.it (L.C.N.); marianomastinu@gmail.com (M.M.); melaniamelis@unica.it (M.M.); 2Department of Otorhinolaryngology, Smell & Taste Clinic, Technical University of Dresden, 01307 Dresden, Germany; anne.wolf4@mailbox.tu-dresden.de; 3Institute of Biochemistry, Food Science and Nutrition, The Hebrew University of Jerusalem, Rehovot 7610001, Israel; tom3rg@gmail.com

**Keywords:** general taste status, taste loss, supervised learning regression, random forest regressor

## Abstract

In healthy humans, taste sensitivity varies widely, influencing food selection and nutritional status. Chemosensory loss has been associated with numerous pathological disorders and pharmacological interventions. Reliable psychophysical methods are crucial for analyzing the taste function during routine clinical assessment. However, in the daily clinical routine, they are often considered too time-consuming. We used a supervised learning (SL) regression method to analyze with high precision the overall taste statuses of healthy controls (HCs) and patients with chemosensory loss, and to characterize the combination of responses that would best predict the overall taste statuses of the subjects in the two groups. The random forest regressor model allowed us to achieve our objective. The analysis of the order of importance of each parameter and their impact on the prediction of the overall taste statuses of the subjects in the two groups showed that salty (low-concentration) and sour (high-concentration) stimuli specifically characterized healthy subjects, while bitter (high-concentration) and astringent (high-concentration) stimuli identified patients with chemosensory loss. Although the present results require confirmation in studies with larger samples, the identification of such distinctions should be of interest to the health system because they may justify the use of specific stimuli during the routine clinical assessments of taste function and thereby reduce time and cost commitments.

## 1. Introduction

There is great physiological variability in taste sensitivity among the healthy human population. This diversity drives food acceptance and selection and affects nutritional status. At the same time, numerous disorders and pharmacological interventions may cause taste dysfunction, i.e., a pathological decrease in sensitivity (hypogeusia), or, in rare cases, complete taste loss (ageusia). Depending on the definition used, hypogeusia can be found in up to 5% of the population aged 5–89 years [1]. The number of individuals with taste disorders increases with age [2,3,4,5,6], reaching up to 15% among the US population aged over 57 years [7]. 

Taste loss diminishes not only the ability to detect noxious or unhealthy substances, but it reduces the joy of consuming tasty foods and, consequently, the pleasures of social eating. Thus, depressive symptoms are commonly detected in patients with chemosensory disorders [8,9]; the frequency of such symptoms in patients suffering from gustatory disorders ranges from 25% to 36% [10,11]. These symptoms become even more frequent in patients with ageusia or qualitative disorders [8].

In healthy adults, another factor associated with a decline in gustatory sensitivity is an increase in body mass index (BMI) [12], and this supports the idea that taste disorders facilitate the consumption of high-calorie foods. Specifically, the threshold for salty taste was higher [13,14], and the ability to identify salty, umami, and bitter tastes was lower [15] in subjects with a high BMI. However, results concerning sweet taste are contradictory: some studies found a direct correlation between a higher glucose sensitivity and a predisposition for developing obesity and diabetes [16,17], while others observed no differences in sweet sensitivity between healthy and obese subjects [18,19]. Importantly, an increased ability to identify sweet taste was found in patients following bariatric surgery in parallel with a significant reduction in weight [20]. 

Sensitivity to the bitterness of 6-n-propylthyouracil (PROP) and phenylthiocarbamide (PTC) also predicts the perception of the other taste qualities [21,22,23], chemical irritants [24], and astringent foods [25,26,27]. For these reasons, it is considered a marker for inter-individual differences in general taste sensitivity. Several studies have reported that individuals with a strong perception of the bitterness of PROP/PTC have a higher sensitivity to other taste stimuli (super-tasters) than individuals with the non-sensitive phenotype (non-tasters) [28,29,30]. PROP sensitivity has also been found to correlate positively with health status [31,32,33,34], but to correlate negatively with BMI [35,36]. Additionally, Essick and coauthors showed that super-tasters had a higher spatial resolution acuity in the tongue than non-tasters using an elegant identification test with 3D-printed letters of the alphabet [37].

Reliable methods for assessing taste perception in patients are crucial in order to define the degree of impairment in patients expressing an altered perception. Most of these methods, although easy to administer, are lengthy procedures in which patients have to maintain focus, and they require a significant commitment from health personnel. Hence, the Seven-iTT was recently proposed for the routine clinical assessment of gustatory and somatosensory functions, including astringency and spiciness [38]. Mastinu and coauthors showed that sweet taste was the taste sensation most frequently identified correctly, followed by salty taste, bitter taste, and sour taste, and that patients with taste impairments had lower identification scores for astringency (a sensation of dryness) and spiciness (a burning sensation). These correlations suggest a connection at a peripheral level between the gustatory and somatosensory perceptions [39,40] that sometimes fail after skull base surgery [41,42]. On the other hand, these results suggest that taste perception, and especially taste dysfunction, are complex and governed by numerous factors.

In the present study, we analyzed with high precision the taste functions of healthy controls (HCs) and patients with chemosensory loss using a supervised learning (SL) approach that provides real-time decision making. We applied the SL regression method using different algorithms. The algorithms were targeted to obtain the most precise predictions of the taste functions of the subjects in the two groups. To this aim, we assessed the intensity ratings of low and high concentrations of each of the six stimuli representative of sweet, sour, salty, bitter, astringent, and spicy tastes. The mean value calculated for each subject was termed the “overall taste status” and was used as the target of the algorithms. As inputs for the algorithms, we used a structured set of data consisting of the sensory, clinical, and anthropometric parameters that had been determined in the subjects of our previous study [38]. We aimed to establish which combination of parameters would best predict the taste function of healthy controls or the taste dysfunction of patients with chemosensory loss. 

## 2. Materials and Methods

### 2.1. Subjects

One-hundred and fifty-three individuals aged 18 to 81 years (38.3 ± 14.3 years; 103 females) were recruited at the Department of Otorhinolaryngology of the TU Dresden from February 2021 to January 2022. Of these, 51 were patients of the Smell and Taste Clinic who self-reported a chemosensory dysfunction. The remaining study sample comprised 102 healthy controls as a reference group. For the purpose of this cross-sectional study, patients were enrolled regardless of the etiology of the taste dysfunction, or their smoking habits. The exclusion criteria were: pregnancy, allergy to substances used in the present study, unmedicated hypo/hyperthyreosis, uncontrolled diabetes mellitus, renal dysfunction, and significant cardiovascular issues. Informed written consent was obtained from all participants prior to their inclusion in the study. The research protocol was approved by the Ethics Review Board at the University Clinic of the Technische Universität Dresden, application number BO-EK-25012021.

### 2.2. Experimental Procedure

The experimental procedure took place in a single session from February 2021 to January 2022. Subjects were requested to abstain from drinking (except water), eating, and chewing gum or using oral care products for at least 1 h prior to testing. Their taste sensitivity of the four primary qualities (sweet, sour, salty, bitter) and sensations of astringency and spiciness were assessed as described below. Before the session, the health statuses of participants were ascertained with detailed medical histories. Weight (kg) and height (m) were recorded in order to calculate the subjects’ BMI (kg/m^2^). Additionally, all participants were screened for depression using the 5-item World Health Organization Well-Being Index (WHO-5) [43,44]. The questionnaire consists of five positively phrased items concerning being in good spirits, feeling relaxed, having energy, waking up fresh and rested, and being interested in things. Each of the five items is rated on a 6-point Likert scale from 0 to 5. The theoretical raw score ranges from 0 to 25. A high score in the WHO-5 indicates a high level of well-being, while a score below 13 indicates poor well-being [43,44]. Subjects were also asked to indicate their preference for consuming spicy foods.

Authors had no access to information that could identify individual participants during or after data collection.

### 2.3. Sensory Measurements

All taste measurements were performed using the same “taste strips” that are used in the validated “Taste Strip Test” (TST, Burghart Company, Wedel, Germany) [45,46]. Taste strips used in the assessments consisted of filter papers impregnated with two concentrations of stimuli representative of four basic taste qualities (sweet, sour, salty, bitter) and of sensations of astringency and spiciness. Taste qualities were presented in a semi-random order, with trigeminal stimuli presented as last due to their persistent sensations. Before every new testing, participants were asked to rinse their mouth with tap water. The evaluation did not include umami sensation due to the low familiarity in the European population [47].

The following two concentrations (one low and one high) for each stimulus were used: 0.4 and 0.05 g/mL sucrose; 0.3 and 0.05 g/mL citric acid; 0.25 and 0.016 g/mL sodium chloride; 0.006 and 0.0004 g/mL quinine hydrochloride; 0.1 and 0.2 g/mL tannin; 2.47 × 10^−5^ and 2.47 × 10^−4^ g/mL capsaicin. After placing each filter paper on the tongue, subjects had to evaluate the perceived intensity for each stimulus using a visual analog scale from 0 to 5 (0 = no taste at all, 5 = extremely strong taste). The ratings of the perceived taste intensity for the two concentrations of each stimulus were called: sweet_low_int and sweet_high_int, sour_low_int and sour_high_int, salty_low_int and salty_high_int, bitter_low_int and bitter_high_int, astring_low_int and astring_high_int, and hot_low_int and hot_high_int. The overall taste status, which is the target of the regressor model, was calculated in each subject as the mean value of intensity ratings for the low and the high concentrations of the six stimuli (score_lowhigh_capsadstr_int1). Subjects of both sexes also had to identify the taste quality of each stimulus by choosing from a list of six possible answers (sweet, sour, salty, bitter, astringency, and hot) in a forced choice procedure [45]. Each correct answer was rated 1, the number of correctly identified tastes was summed up in the total taste score (score_lowhigh_capsadstr1) whose maximum value was 12. The scores of the correct answer for the two concentrations of each stimulus were labeled as follows: salty_low_taste_correct; salty_high_taste_correct; sweet_low_taste_correct; sweet_high_taste_correct; sour_low_taste_correct; sour_high_taste_correct; bitter_low_taste_correct; bitter_high_taste_correct; hot_low_taste_correct; hot_high_taste_correct; astring_low_taste_correct; and astring_high_taste_correct. 

The complete procedure required 20 min for each subject. 

### 2.4. Supervised Learning

The automatic prediction of the overall taste status of healthy controls and patients with chemosensory loss was carried out from January to October 2022 by SL algorithms exploiting the subjects’ parameters that were presented in the data model as predictive variables (features). We used the SL regressors because they are specific to predict continuous outcomes such as the overall taste status of subjects [48]. The SL regressors learned and created automatic regressor models that assessed between-subjects differences and returned, with high accuracy, a value prediction of the overall taste status in healthy controls or patients with chemosensory loss. The following algorithms were used: logistic regression, random forest regressor, and CatBoost regressor. During training, the different algorithms learn the hidden patterns in the structured dataset and then take new unforeseen inputs (test dataset) to predict the target value (the overall taste status). In order to analyze the overall taste status of healthy controls and patients with chemosensory loss and individuate specific stimuli or their combination that can best predict the overall taste status of the two groups, it was necessary to apply the regressor model separately to the dataset of healthy controls and patients with chemosensory loss. Therefore, we performed two SL regressor experiments: experiment 1 included healthy controls (*n* = 102) and experiment 2 included patients with chemosensory loss (*n* = 51). 

The interpretation of the results of the random forest regressor model has been performed by using Shapley additive explanations (SHAPs) [49], which is a game-theoretic method that allows us to link the importance of each feature with its effect. SHAPs return a specific plot for each random forest regressor experiment representing the impact of each feature in that experiment. 

The following data processing operations, which are a crucial phase in the performance of an SL project, were applied:The correlations between numerical parameters and those between each numerical feature and the target, which were a fundamental step to understand the data structure, were also considered to include a feature in the dataset (Figure 1).The choice of features that are used by the algorithm as predictive variables of the target: The set of parameters most suitable for our case study was selected as features by expert researchers in taste physiology and an ML engineering, based on their domain knowledge, from a database of the sensory, clinical, and demographic parameters [50]. In addition, since two features strongly correlated with each other have almost the same effect on the dependent variable, one of them was dropped to reduce the noise that could impact algorithm performance [51]. Specifically, the sum of the scores of the correct answer for the two concentrations of salty, sweet, sour, bitter, hot, and astringency; the sum of the scores of the correct answers for the two concentrations of salty, sweet, sour, and bitter; and the intensity ratings for the two concentrations of salty, sweet, sour, and bitter strongly correlated with each other. The first summated variable that included evaluations of all stimuli was selected, and the latter two were excluded.Handling of missing values: Every line of the subject that represents lacking information in some column was eliminated.Elimination of duplicate values: In fifty-eight subjects of the control group, all sensory measurements were repeated twice. The column relative to these repeated measures, and those of the overall taste status and total taste score calculated in these subjects (twenty-nine columns in total) were eliminated from the dataset.Converting the dataset content into a language that an algorithm can understand: This included the one hot encoding, which encodes categorical data into numerical data and the normalization of the numerical data, which consist of transforming a real range of numerical values in a range between 0 and 1.

After data processing operations that included the remotion of all non-significant and correlated features, we increased the parameters of the numbers of estimators and the maximum of depth of the SL regressors. Moreover, we used 3-fold cross-validation, which mixes and splits data into two groups (training data, 66.66% and test data, 33.33%) three times using different subsets of data each time.

The evaluation of the performance of the algorithms was found by metrics, such as mean absolute percentage error (MAPE) and mean squared error (MSE), that assess the differences between the observed and predicted values. In particular, MAPE represents the error percentage of predicted values, while MSE represents the average of the summation of the squared difference between the actual output value and the predicted output value. The overall behavior of our regressors was subsequently evaluated by the automatic optimization of their hyperparameters by grid search algorithm. 

### 2.5. Statistical Analysis

Fisher’s exact test was used to analyze differences between healthy controls and patients with chemosensory loss regarding the frequency of correct answers for the two concentrations of each stimulus, gender, and preference for spicy foods. *T* test was used to compare differences between healthy controls and patients with chemosensory loss in age, BMI, depression, taste intensity ratings for each stimulus, and total taste score. Statistical analyses were conducted using STATISTICA for WINDOWS (version 7; StatSoft Inc, Tulsa, OK, USA). *p* values < 0.05 were considered significant. *p* values of T test were adjusted by Bonferroni correction (adjusted *p* = *p*/number of groups being compared) (*p* values < 0.0031 were considered significant).

## 3. Results

Mean values ± SD or frequencies of the sensory, clinical, and anthropometric parameters determined in healthy controls and patients with chemosensory loss are shown in Table 1. The *T* test adjusted by Bonferroni correction showed that ratings of the taste intensity in response to low concentrations of each stimulus and high concentrations of sweet, sour, and astringency as well as the total taste score of healthy controls were higher than those of patients with chemosensory loss (*p* ≤ 0.0006). The depression level was higher in patients with chemosensory loss than in healthy controls (*p* < 0.0001). The number of correct answers for the low concentrations of salty, bitter, hot, and astringency and the high concentrations of sweet, sour, and astringency alongside the number of subjects who enjoy spicy foods were higher in healthy controls than in patients with chemosensory loss (χ2 ≥ 4.79; *p* ≤ 0.041; Fisher’s test). No differences in the ratings of taste intensity in response to high concentrations of salty, bitter, and hot, age, BMI, gender, or in the number of correct answers for low concentrations of sweet and sour, and high concentrations of salty, bitter, and hot were found between the two groups (*p* > 0.05).

The metrics of evaluation of the performance of the algorithms, MAPE and MSE, which measure the accuracy of a forecast system, showed that the random forest regressor was the best algorithm to predict the values of the overall taste status with high precision. The scatterplots showing experimental values vs. predicted values of the overall taste status obtained with the random forest regressor in healthy controls, and patients with chemosensory loss are shown in Figure 2. The values of MSE, which evaluate how estimated values are close to experimental values, were 0.019 and 0.014 in the two groups. Furthermore, the MAPE values, which represent the error percentage of predicted values, were 4.55% and 8.40% in healthy controls and patients with chemosensory loss, respectively. 

The random forest regressor allowed us to determine the order of importance and the contribution of the sensory, clinical, and anthropometric features on the prediction of the overall taste status in the two groups, and the interpretation by SHAP algorithm allowed us to obtain an overview of their impact on the prediction. 

Specifically, the rating of the perceived taste intensity for low concentrations of salt was the most important feature for model learning in healthy controls, and its contribution on the prediction of the overall taste status, estimated as the average impact on the model, was 0.079 (Figure 3A). This feature was followed in order of importance from second to tenth by: the intensity rating for high concentrations of sour, for low concentrations of bitter, for high concentrations of sweet, for high concentrations of bitter, for low concentrations of hot, for high concentrations of salty, for low concentrations of astringency, the total taste score, and the intensity rating for low concentrations of sweet. It is interesting to note that depression status was a significant feature (sixteen in importance order). These features had an average impact on the model lower than 0.057. The link between importance and the effect of the features on the overall status of healthy controls is shown in the SHAP summary plot (Figure 3B). The plot highlights that high estimated values (pink) of the perceived taste intensity rating for low concentrations of salt and high concentrations of sour had a strong and positive impact to make a prediction of high values of the overall taste status, and the low estimated values (blue) of these features strongly pushed the model prediction toward low values of the overall taste status. The mean values of the positive impacts of these two features were 0.052 and 0.040, while those of the negative impacts were −0.155 and −0.106, respectively. High estimated values (pink) of the successive eight features had a positive impact (≤0.057) to make a prediction of high values of the overall taste status; medium values (violet) impacted the model to make a prediction of medium values of the overall taste status; and low estimated values (blue) of these features pushed the model prediction toward low values of the overall taste status (negative impact ≤ −0.069).

The importance of features in facilitating the learning of the random forest regressor to predict the overall taste status, and an overview of how the most important features impact it to make a prediction in patients with chemosensory loss are shown in Figure 4. In this case, the ratings of the perceived taste intensity for high concentrations of bitter and for high concentrations of astringency were the first two most important features for the model to predict the overall taste status of this group, resulting in a similar contribution on the model prediction of the overall taste status, whereby the average impacts were 0.105 and 0.100, respectively (Figure 4A). These were followed in order of importance from the third to the tenth by: the total taste status, the intensity rating for low concentrations of salty, for high concentrations of sweet, for low concentrations of bitter, for low concentrations of sweet, for high concentrations of sour, for high concentrations of salty and for low concentrations of sour. These features had an average impact on the model lower than 0.077. The SHAP summary plot highlights that high estimated values (pink) of the rating of the perceived taste intensity for high concentrations of bitter and for high concentrations of astringency had a strong and positive impact to make a prediction of high values of the overall taste status, and the low estimated values (blue) of these features strongly pushed the model prediction toward low values of the overall taste status (Figure 4B). The mean values of the positive impacts of these two features were 0.085 and 0.096, while those of the negative impacts were −0.134 and −0.105, respectively. High estimated values (pink) of the successive eight features had a positive impact (≤0.098) to make a prediction of high values of the overall taste status; medium (violet) and low (blue) estimated values of these features impacted the model to make a prediction of medium and low values of the overall taste status (negative impact ≤ −0.105).

The error percentage of predicted values of the overall taste status (MAPE values) calculated in the datasets of the two groups by using as inputs for the algorithm only the most important feature for the two groups (the intensity rating for low concentrations of salt in healthy subjects and high concentrations of bitter in patients), was 12.35% and 17.94% in healthy controls and patients with chemosensory loss, respectively. When the second important features were also included in the model (high concentrations of sour for healthy subjects and high concentrations of astringency for patients), the error percentage of predicted values of the overall taste status was 11.05% and 17.16% in healthy controls and patients with chemosensory loss, respectively. 

## 4. Discussion

This study used the SL regression method to establish which combination of sensory, clinical, and anthropometric parameters best predicts the overall taste status of healthy controls or that of patients with chemosensory loss. The subjects included as patients with chemosensory loss were selected based on what they reported. The values of the sensory parameters, which were used as predictive variables of the overall taste status, as well as the correlation between them and the target, confirm that they showed a lower taste sensitivity compared to the subjects of the control group. The low taste sensitivity of these subjects was linked to their higher depression status according to what was already reported [8].

The random forest regressor was the best model to deeply understand differences among subjects and obtain, with high precision, the value of the overall taste status of subjects. Moreover, the random forest regressor allowed us to establish the impact of each parameter based on prediction, identifying the combination of the biological parameters that could best predict the overall taste status of healthy subjects or that of patients with chemosensory loss. The performance of our approach was tested by the metrics of evaluation, MAPE and MSE, which allowed us to verify that the estimated values by the random forest regressor were strictly close to experimental values with an error percentage of 4.55% and 8.40% in healthy controls and patients with chemosensory loss, respectively. In addition to these advantages of the SL method adopted, which are further outlined below, it is necessary to point out some disadvantages or limitations of this approach. First, it is known that machine learning methods need big data to fit the algorithms, and bias is expected to be larger for smaller datasets. Although dataset size is not necessarily a barrier to a high-performing model as shown by the metrics of evaluation, our results were obtained in small-sized datasets, mainly that of patients. Therefore, it would be useful to confirm our results in larger datasets. Second, the interpretation of the results of the regressor model used has been performed by using the SHAP algorithm, which allows us to link the importance of each feature with its impact on the prediction. Although SHAP is the best algorithm to achieve this goal, other algorithms can also be used [52]. These two limitations of the study need to be further explored and will be the topic of future research projects.

The random forest regressor also allowed us to achieve the order of importance of each feature on the prediction of the overall taste status of the subjects of the two groups. The regressor identified the rating of the perceived taste intensity for low concentrations of salt to be the most important parameter in healthy subjects, while the rating of the perceived taste intensity for high concentrations of bitter was the most significant in patients with chemosensory loss. Moreover, the interpretation of the link between the importance and the effect of parameters showed that high estimated values of the perceived intensity for low concentrations of salt had a strong and positive impact to predict high values of the overall taste status in healthy subjects. On the other hand, high estimated values of the rating of the perceived taste intensity for high concentrations of bitter had a strong and positive impact to predict high values of the overall taste status in patients with chemosensory loss. Low values of these two parameters strongly pushed the model prediction toward low values of the overall taste status in both groups. The fact that the most important parameter to predict the overall taste status in healthy controls was the rating of the perceived intensity for low salt concentrations is not surprising since, as we can see from experimental values shown in Table 1, low concentrations of salt evoked the highest response compared to low concentrations of other stimuli, and a high percentage of subjects (85.3%) recognized it correctly. The rating of the perceived taste intensity for high concentrations of bitter, which was the most important feature in patients with chemosensory loss, was correctly recognized by 76.5% of patients. The bitter taste was also an important stimulus in healthy controls, ranking third in importance order at low concentrations, and was correctly recognized by 61.8% of subjects. The difference in the importance of bitter taste that we found in the two groups may be explained by possible genetic differences between the two groups. In our pilot study, we found that, unlike healthy controls, in patients with taste disorders, a taster haplotype in the gene coding for bitter TAS2R38 receptor is not sufficient to exhibit high responses, suggesting that the genetic constitution may represent a risk factor for the development of taste disorders [53]. Furthermore, based on the findings which suggest that the TAS2R38 pathway is an immune response target [33,54], patients with the non-taster haplotype, who show lower responsiveness, could have a higher susceptibility to oropharyngeal infections, which may contribute to their chemosensory loss. 

The second most important parameter in learning the model in healthy controls was the perceived intensity rating for high concentrations of sour, which was the stimulus that evoked the highest response and which 91.2% of subjects correctly recognized. High estimated values had a positive impact to predict high values of the overall taste status, while low estimated values strongly pushed the model prediction toward low values of the overall taste status. Less important was the impact that sour had on the prediction of the overall taste status in patients with chemosensory loss, in which it was the eighth feature in order of importance, suggesting that the impact of sour on taste function may be more relevant in subjects that have no pathologies that cause chemosensory loss. 

The second most important parameter in the model’s learning in the patients with chemosensory loss was the perceived intensity rating for high concentrations of astringency, which strongly impacted the prediction of the overall status of this group in a way equivalent to that which was performed by the most important feature, high concentrations of bitter (the values of their contribution to the model prediction were 0.105 and 0.100, respectively). Less important was the contribution and impact that astringency (eighth in importance order at low concentrations and eleventh at high concentrations) had on the prediction of the overall taste status in healthy subjects, in which low estimated values pushed the model prediction toward low values of the target. These results allow us to speculate that the impact of astringency on taste function could be more important in subjects with low sensitivity. This result deserves to be further investigated in future studies.

It is interesting to note that the performance of our model applied in the datasets of the healthy controls and that of patients, by using as inputs for the algorithm only the most important features for the two groups, allowed us to verify that the estimated values were strictly close to experimental values with an error percentage of 12.35% and 17.95%, respectively. By also including in the model the second important features as the predictive variables, the error percentages of the predicted values of the overall taste status decreased of 1.3% and 0.79%, in healthy controls and patients, respectively. The low error percentage of the predicted values that we found by including in the model low concentrations of salt and high concentrations of sour for healthy subjects, and high concentrations of bitter and high concentrations of astringency stimuli for patients, offers the great advantage of having identified the two most important stimuli that can predict the overall taste status in the two groups. This is extremely important because it allows for the reduction in the experimentation times from 20 min to 3 min for each subject, with respect to the complete procedure of the “12-taste strip test”, which lasts 20 min. 

## 5. Conclusions

In conclusion, our results indicated that the random forest regressor is a reliable strategy to analyze taste function by exploiting a structured dataset consisting of sensory, clinical, and anthropometric parameters previously determined in the participants [38] as inputs for the algorithm. Furthermore, the proposed approach, which provides real-time decision-making, allowed us to identify with high-precision different stimuli and their combination that can best predict the overall taste status in the two groups. The low concentrations of salt and high concentrations of sour were specific for healthy subjects, while the high concentrations of bitter and high concentrations of astringency stimuli were the most indicative ones for pathological taste disorders. These four stimuli strongly impacted the model prediction mostly in the subjects where they evoked low responses (the impact values increased two-fold with respect to those of the other stimuli), suggesting that a low response to these stimuli characterizes taste dysfunction. However, the present results need confirmation in a larger study.

It is well known that healthy humans exhibit considerable physiological variation in taste sensitivity. However, a significant loss of taste, such as hypogeusia or ageusia, has been linked to a variety of pathological conditions or pharmacological treatments. When conducting routine clinical assessments to analyze taste function, solid psychophysical techniques are essential tools. However, these are drawn-out processes that demand a significant commitment both by patients and healthcare professionals. In an effort to gain a deeper understanding of taste loss, we developed the SL regression approach that allows us to automatically and accurately compare the general taste status of healthy controls and patients with chemosensory loss and identify with high precision the different stimuli and their combinations, which can most accurately predict the general taste status in the two groups. The identification of these differences is of great interest to the health system because they allow the use of specific stimuli during the routine clinical measurements of taste function, reducing commitment in terms of time and costs. The innovation introduced by our approach and its significance for the advancements in the field will be able to simplify testing, especially during routine clinical examinations (e.g., annual physical exams) or in follow-up visits.

## Figures and Tables

**Figure 1 biomedicines-11-02133-f001:**
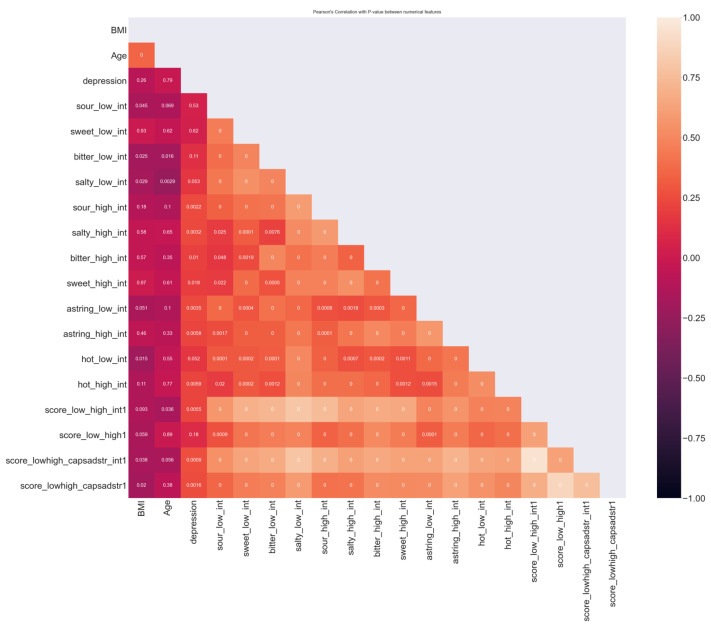
Linear correlation analysis between the numerical features of the dataset and those between numerical features and the target. The bar color on the right side on the Y axis represents the value of linear correlation between features. *p* values are indicated inside each square and derived from Pearson’s coefficient analysis.

**Figure 2 biomedicines-11-02133-f002:**
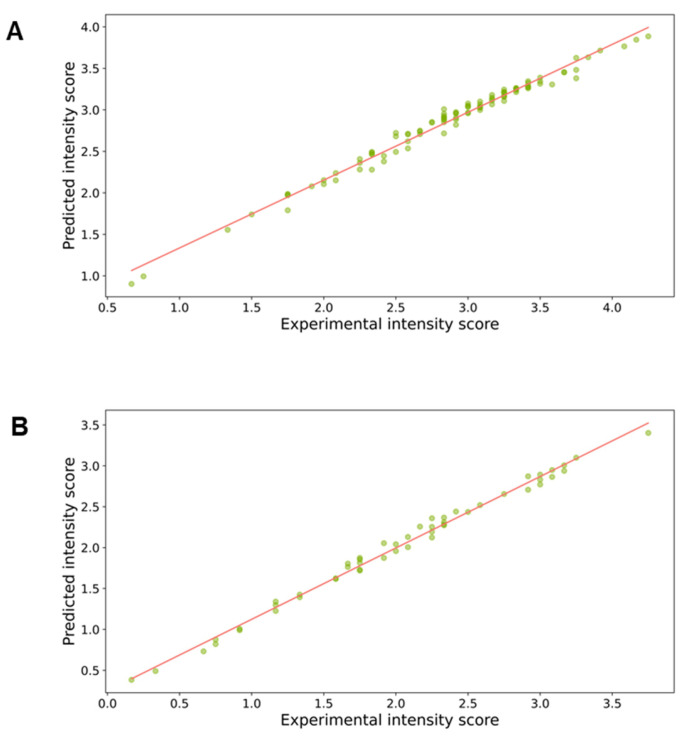
Scatterplots of experimental values vs. predicted values (dots) of the overall status obtained with the random forest regressor in healthy controls (*n* = 102) (**A**) and in patients with chemosensory loss (*n* = 51) (**B**). The lines represent the relationships between the predictive and experimental (independent and dependent) variables.

**Figure 3 biomedicines-11-02133-f003:**
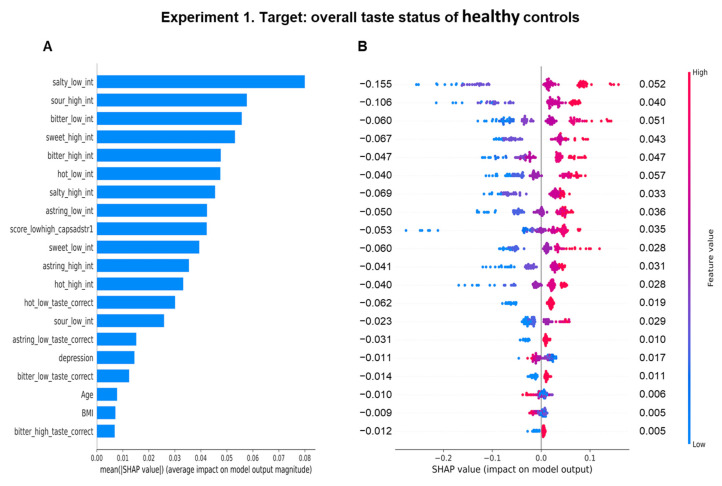
Importance and impact of the sensory, clinical, and anthropometric features on the overall taste status prediction determined with random forest regressor in healthy controls. Importance of features in the learning of the model to understand the overall taste status (**A**). The Y axis represents the order of the importance of features, while the average impact on the model output is represented on the X axis. The SHAP summary plot in the healthy controls (**B**). The left-hand side of the Y axis represents the descending order of importance; the X axis represents the SHAP value, i.e., the impact on the output model. The color represents the feature value: high values have a pink color, while low values have a blue one. Numbers indicated for each line represent the mean of the positive impact values and the negative impact of each feature.

**Figure 4 biomedicines-11-02133-f004:**
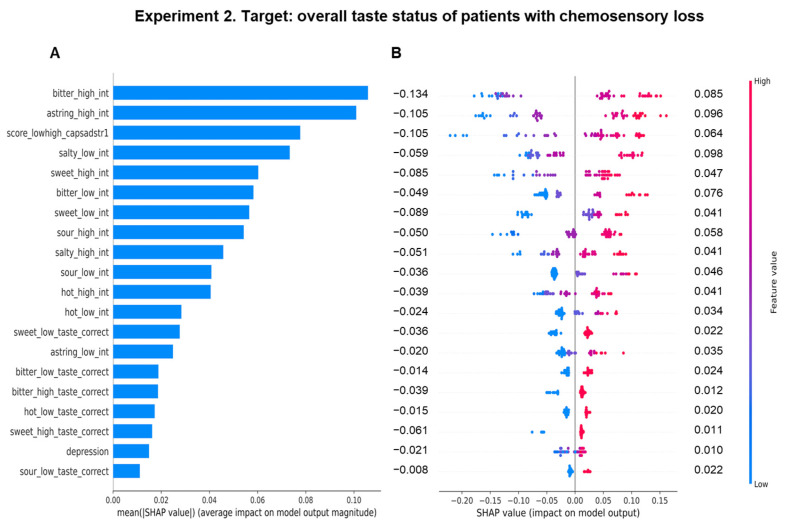
Importance and impact of the sensory, clinical, and anthropometric features on the overall taste status prediction determined with random forest regressor in patients with chemosensory loss. Importance of features in the training model to understand the overall taste status (**A**). The Y axis represents the order of importance of the features, while the average impact on the model output is represented on the X axis. The SHAP summary plot in patients with chemosensory loss (**B**). The left-hand side of the Y axis represents the descending order of importance; the X axis represents the SHAP value, i.e., the impact on the output model. The color represents the feature value: high values have a pink color, while low values have a blue one. Numbers indicated for each line represent the mean of the positive impact values and the negative impact of each feature.

**Table 1 biomedicines-11-02133-t001:** Sensory, clinical, and anthropometric parameters of healthy controls and patients with chemosensory loss.

Features	Healthy Controls (*n* = 102)	Patients with Chemosensory Loss (*n* = 51)	*p*-Value
Numerical type			
astring_low_int	2.24 ± 1.32	0.84 ± 1.10 *	<0.0001
bitter_low_int	2.40 ± 1.49	1.06 ± 1.26 *	<0.0001
hot_low_int	2.04 ± 1.30	0.84 ± 1.24 *	<0.0001
salty_low_int	3.03 ± 1.07	1.92 ± 1.18 *	<0.0001
astring_high_int	3.53 ± 1.09	2.43 ± 1.43 *	<0.0001
score_lowhigh_capsadstr1	9.37 ± 1.92	7.18 ± 2.47 *	<0.0001
sour_high_int	3.90 ± 0.85	3.27 ± 1.05 *	0.0001
sweet_low_int	1.99 ± 1.18	1.22 ± 1.14 *	0.0002
sweet_high_int	3.58 ± 0.89	2.90 ± 1.38 *	0.0003
sour low int	1.37 ± 1.22	0.68 ± 0.97 *	0.0006
bitter_high_int	3.15 ± 1.43	2.57 ± 1.43	0.018
hot_high_int	3.54 ± 1.16	3.04 ± 1.33	0.018
salty_high_int	3.79 ± 0.89	3.45 ± 1.27	0.054
depression	17.15 ± 3.62	14.14 ± 5.18 *	<0.0001
BMI (kg/m^2^)	23.66 ± 4.02	25.61 ± 5.55	0.014
Age (y)	36.70 ± 14.43	40.45 ± 12.62	0.116
Categorial type			
astring_low_taste_correct/non (*n*)	78/24	15/36 #	<0.0001
astring_high_taste_correct/non (*n*)	86/16	24/27 #	<0.0001
hot_low_taste_correct/non (*n*)	77/25	18/33 #	<0.0001
salty_low_taste_correct/non (*n*)	87/15	33/18 #	0.0039
sour_high_taste_correct/non (*n*)	93/9	36/15 #	0.0014
bitter_low_taste_correct/non (*n*)	63/39	22/29 #	0.0221
sweet_high_taste_correct/non (*n*)	100/2	46/5 #	0.0415
sour_low_taste_correct/non (*n*)	32/70	10/41	0.0877
sweet_low_taste_correct/non (*n*)	72/30	32/19	0.2122
hot_high_taste_correct/non (*n*)	97/5	46/5	0.2061
bitter_high_taste_correct/non (*n*)	77/25	39/12	0.531
salty_high_taste_correct/non (*n*)	90/12	45/6	0.596
Like_to_eat_spicy/non (*n*)	66/36	23/28 #	0.016
sex_f1_m2 (women/men; *n*)	65/37	36/15	0.255

Values are means ± SD, or number of correct answers, or number of subjects. Body mass index, BMI; significant differences in mean values between healthy controls and patients with chemosensory loss are indicated by * (*p* ≤ 0.0006; T test adjusted by Bonferroni correction), while differences in frequency distribution are indicated by # (*p* < 0.041; Fisher’s method).

## Data Availability

All datasets and code presented in this study are freely available in Github at the link: https://github.com/lala-sudo/SCORE_LOW_HIGH_CAPSADSTR_INTENSITYW (accessed on 29 May 2023).

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
