# Peer review of "A Supervised Learning Regression Method for the Analysis of the Taste Functions of Healthy Controls and Patients with Chemosensory Loss"

_biomedicines, 2023, doi:10.3390/biomedicines11082133_

Round 1

Reviewer 1 Report

This manuscript was entitled as “ A supervised learning regression method for the analysis of 2 taste function of healthy controls (HC) and patients with chemosensory loss”. The authors concluded that a supervised learning (SL) method could predict the overall taste function with high precision. However, the authors might need to explain some characteristics of this method in order to prove its clinical applicability.

1.         What are the advantages, disadvantages and limitations of SL method used to predict the overall taste function of HC and patients with chemosensory loss ?

2.         When 12 taste strips were used to evaluate the taste function of HC and patients with chemosensory loss in SL method, what are the advantages and disadvantages of SL method to predict the overall taste function of HC and patients with chemosensory loss as compared with 12 taste strip method to measure the taste function of HC and patients with chemosensory loss?

3.         What are the indications of the SL method used to predict the overall taste function of HC and patients with chemosensory loss in the clinical practice?

4.         How does the SL method predict a subject to suffered from taste dysfunction in the clinical practice? Does the SL method provide a numerical result? Can the SL method predict whether the taste function improves or worsens in the follow-up visit visits?

Author Response

We have reworked the manuscript according to the Reviewers’ comments and suggestions.

In the revised manuscript the changes made according to Reviewer 1 suggestions are highlighted in red, those according to Reviewer 2 in blue.

Reviewer #1

Comments and Suggestions for Authors

This manuscript was entitled as “ A supervised learning regression method for the analysis of 2 taste function of healthy controls (HC) and patients with chemosensory loss”. The authors concluded that a supervised learning (SL) method could predict the overall taste function with high precision. However, the authors might need to explain some characteristics of this method in order to prove its clinical applicability.

  1. What are the advantages, disadvantages and limitations of SL method used to predict the overall taste function of HC and patients with chemosensory loss?

 Reply: we comply with the Reviewer’s request. We added in the Discussion section the disadvantages and limitations of SL method used (lines 372-384): “ In addition to these advantages of the SL method used, which are further outlined below, it is necessary to point out some disadvantages or limitations of this approach. First, it is known that Machine Learning methods need big data to fit the algorithms, and bias is expected to be larger for smaller datasets. Although dataset size is not necessarily a barrier to a high-performing model as shown by the metrics of evaluation, our results were obtained in datasets with low size, mostly that of patients. Therefore, it would be useful to confirm our results in larger datasets. Second, the interpretation of the results of the regressor model used has been performed by using SHAP algorithm which allows us to link the importance of each feature with its impact on the prediction. Although, SHAP is the best algorithm to archive this goal [52], also other algorithms can be used [53]. These two limitations of the study need to be explored more deeply and will be the topic of future research projects”.

We also added a sentence to the abstract and the conclusion saying that the present results need confirmation in larger samples (Line 25): “…Although the present results require confirmation in studies with larger samples…” and lines 462-463: “…However, the present results need confirmation in a larger study…”

In addition, we point out to the Reviewer that in lines 385-435, the differences of impact of the more important two features for the two groups on the prediction of the overall taste status of HC and patients are highlighted.

  1. When 12 taste strips were used to evaluate the taste function of HC and patients with chemosensory loss in SL method, what are the advantages and disadvantages of SL method to predict the overall taste function of HC and patients with chemosensory loss as compared with 12 taste strip method to measure the taste function of HC and patients with chemosensory loss?

 Reply: We added in the Discussion section the following sentences (lines 442-449) to explain the advantage of SL method with respect to the “12 taste strip method” to measure the taste function of HC and patients: ”The low error percentage of the predicted values that we found by including in the model the low concentration of salt and high concentration of sour for healthy subjects and the high concentration of bitter and high concentration of astringency stimuli for patients, offers the great advantage of having identified the two most important stimuli that can predict the overall taste status in the two groups. This is extremely important because permits the reduction of the experimentation times from 20 min to 3 min for each subject, with respect to the complete procedure of the “12-taste strip test” that has a duration of 20 min”.

We also added to “Sensory measurements paragraph” (line 156) the flowing sentence: ” The complete procedure required 20 min for each subject.”

  1. What are the indications of the SL method used to predict the overall taste function of HC and patients with chemosensory loss in the clinical practice?

Reply: we comply with the Reviewer’s request. We added the following sentences (lines 464-479): “It is well known that healthy humans exhibit considerable physiological variation in taste sensitivity. However, significant loss of taste, such as hypogeusia or ageusia, has been linked to a variety of pathological conditions or pharmacological treatments. When conducting routine clinical assessments to analyze taste function, solid psychophysical techniques are essential tools. However, these are drawn-out processes that demand a significant commitment from both patients and healthcare professionals. In an effort to gain a deeper understanding of taste loss, we developed the SL regression approach that allows us to automatically and accurately compare the general taste status of healthy controls and patients with chemosensory loss and identify with high precision different stimuli and their combination that can most accurately predict the general taste status in the two groups. The identification of these differences is of great interest to the health system because they allow the use of specific stimuli during the routine clinical measurements of taste function reducing the commitment in terms of time and costs. The innovation introduced by our approach and its significance for the advancements in the field will be able to simplify testing especially during routine clinical examinations (e.g., annual physical exams) or in the follow-up visits.”

 4. How does the SL method predict a subject to suffered from taste dysfunction in the clinical practice? Does the SL method provide a numerical result? Can the SL method predict whether the taste function improves or worsens in the follow-up visit visits?

 The study proposed by the reviewer is extremely interesting and will be topic of our next research project, however, it goes beyond the scope of the present study.

We would like to point out that our objective was to obtain the most precise prediction on the taste function of HC and patients and individuate specific stimulus or their combination that best can predict the overall taste status (a continuous target) of HC and patients with chemosensory loss. To archive this objective, we choose two groups of subjects, healthy and patients of the Smell and Taste Clinic who self-reported a chemosensory dysfunction. Therefore, our method cannot predict if a subject suffers from taste dysfunction. To archive this would be necessary to use a classifier algorithm that cannot be used to predict continuous outcomes, such as the overall taste status. Differently, to obtain the most precise prediction of the overall taste status and individuate specific stimuli that best can predict them in the two groups, was necessary to apply a regressor model (specific to predict continuous outcomes) separately to the datasets of HC and patients with chemosensory loss.

Regarding the question if the SL method can predict whether the taste function improves or worsens in the follow-up visit visits, please see the previous reply.

Reviewer 2 Report

This is a very interesting and important paper to the scientists in the field. There are different types in the sense of taste and, taste dysfunction does not happen to all the different types of taste sense in the same way. The results of this study will be a great help to the clinicians who diagnose taste dysfunction. I highly recommend the publication of this paper.

Author Response

Reviewer #2

Reply: We thank the reviewer; we really appreciate the Reviewer's nice words about our paper

Reviewer 3 Report

In this manuscript, the authors used the supervised learning regression method to analyze the overall taste status of healthy and patients with chemosensory loss and to characterize the combination of responses that best can predict the overall taste status of two groups. The Random Forest regressor was the method that achieved best results. The analysis of the results achieved showed, for the two groups, that salty (low concentration) and sour (high concentration) stimuli specifically characterized healthy subjects, while bitter (high concentration) and astringent (high concentration) stimuli identified patients with chemosensory loss. The identification of these distinctions is of interest to the health system since they may allow the use of specific stimuli during routine clinical assessments of taste function reducing the commitment in terms of time and costs.

Globally, the manuscript is very well written and organized. In fact, there are only a very small number of typing/editing corrections that I recommend; please refer to the attached commented document, where these corrections are highlighted.

Please refer to the comments above.

Author Response

We really appreciate the Reviewer's nice words about our paper. In the revised manuscript the corrections according to the recommendations of Reviewer 2 are highlighted in blue.